# Teaching and Investigating on Modelling through Analogy in Primary School

Leonardo Colletti [1,*] , Soufiane Krik [2] , Paolo Lugli [2] and Federico Corni [1]

1    Faculty of Education, Free University of Bozen-Bolzano, viale Ratisbona, 16, 39042 Bressanone, Italy; federico.corni@unibz.it
2    Faculty of Engineering, Free University of Bozen-Bolzano, piazza Domenicani, 3, 39100 Bolzano, Italy; soufiane.krik@unibz.it (S.K.); paolo.lugli@unibz.it (P.L.)
*    Correspondence: leonardo.colletti@unibz.it

**Abstract:** Physics deals with complex systems by reducing them to relationships between a limited number of relevant quantities and general principles. Since we live in a reality characterised by an increasing complexity in all fields, an indispensable challenge arises for education to turn basic science instruction into a key stage of education per se. Is it possible to introduce some aspects of the physics approach as early as the first school years? Which ones, how, and with what results? Here, results of the initial phase of a three-year project on complexity are presented. This educational innovation path has been developed for elementary and middle schools and is designed as a gentle introduction to complex and systemic thinking. It aims to foster in children reasoning by analogies and the development of simple but effective and versatile basic concepts. The project exploits the use of the small set of primary metaphors already available in children's cognitive toolkit to apply them to describing the characteristics of various circuits, from marbles to water and air to electricity. Pupils' feedback was analysed through a single case study with a qualitative and quantitative methodology. Results were encouragingly positive and showed a wide range of abilities to capture and develop analogies on the topic of the circuit.

**Keywords:** physics in primary schools; electric circuits; analogies and abstraction; embodied mind; imaginative education





## 1. Introduction

An important task of school education is to provide the young learner's mind with the tools to deal effectively with the multiple aspects under which reality presents itself. This also means being able to foster the ability to extract meaningful conceptual hierarchies from a world that itself appears increasingly complex and interconnected, whether one is dealing with an electronic device, a mechanical ensemble, a biological system, or a socio-economic context.

For example, physics—the quintessential model of the scientific method—has proved particularly effective in devising recurring and versatile patterns of explanation that are powerful in bringing out the aspects that truly play a major role in a system, from the atomic nucleus to stellar superclusters, i.e., over a wide range of dimensional and energy scales.

In this research, we set out to question whether and how it is possible to teach the very fundamentals of the physics approach—rather than its results—as early as the first years of education. Indeed, we would like to focus on physics as a masterful method of conceptually organising the world rather than as a list of notions to be assimilated. Certainly, it is possible to question at the scholarly level what the essential identity of the method of physics is, but here, not intending to devote ourselves to that topic, we refer to those aspects which are the most widely acknowledged by the scientific community, namely, the use of causal reasoning, the creation of simplified abstract models, and the

experimental control through observation and measurement, supported by studies such as that of Osborne et al. [1].

Is it therefore possible to guide children into identifying the variables that drive the causal connection in a process and into recognising the similarities between supposedly different situations? To form a mental model for a system or class of systems?

Let us consider some topics that are common to the primary school curriculum: The food cycle, the water cycle, and electric circuits. Through the school years, they are taught at different times and with different styles and levels of depth. They concern different phenomena and fields of knowledge. At the same time, children also experience other activities, in and out of school, such as the daily routine of leaving home in the morning, full of energy, and returning later, having expended the energy in different places and activities; or gymnastic trails in the gym, or, particularly in our Alpine region, downhill skiing, where a skilift takes them to the top of the mountain from which they can then descend to their starting point. Beyond the obvious differences between all these situations, the trained eye can recognise a common basic structure, namely, that of the *circuit*: an agent travels along a path performing a certain task on one or more patients and finally returning to the starting point, where it can recharge.

This recognition exemplifies an attitude typical of science, i.e., that of being able to reduce the complexity of a system (or process) to the variables that really matter in it, while at the same time finding similarities between systems (and processes) that are apparently even very dissimilar to one another. In order for children to grow and gradually approach this way of engaging effectively and synthetically with reality, it is necessary to offer them opportunities that can explicitly lead them to increasingly refined levels of abstraction.

The AT-NE-ST three-year project (the acronym means *Discovering complexity: Advanced technologies for an education in storytelling and systemic thinking*) at the Free University of Bozen-Bolzano aims at bringing K-8 students (i.e., kindergarten, primary, and middle school pupils) closer to the theme of extrapolating meaning from the complexity of data provided by sense experience.

Complexity is a diverse and partly ambiguous field. Often, complex means something extremely difficult, at other times the emphasis is on the enormity of a data set, and at other times on the high degree of interconnections between different parts of a system or many systems. As a subject of physics, complexity refers typically to topics such as the three-body problem, deterministic chaos, neural networks, and ecosystems. In this paper, however, by complexity, we do not relate to a form of content but to a methodological approach that underpins all physics. In the first phase of the project—outlined here and aimed at elementary school pupils up to grade three—complexity is understood indeed as the multiplicity of characteristics with which the events and phenomena of nature present themselves to us, but can nevertheless be reduced to a unifying core, particularly through the use of analogy and the development of synthetic concepts. This unifying core thus consists of the set of traits common to the set of phenomena in question: if at first we are faced with many different phenomena and variables, through this operation of weaving analogies, we are eventually faced with a single entity (which we cannot call anymore a phenomenon—because it does not appear to our senses—but is evident to our minds).

For this purpose, an activity based upon observation and hands-on practice was designed and experimented to lead pupils to identify similarities between circuits of different natures, such as mechanical circuits where marbles, air, or water set a wheel in rotation, and simple electric circuits, in which a battery drives the electric current that lights a light bulb.

The teaching of electricity has been widely addressed, at different levels and with different methodologies, but it has mainly concerned middle and high school teaching, often on the topic of electric circuits [2–10], but also on other topics such as electric transport in solids [11,12] or electrostatics [13,14]. With regard to K-5 children, investigations on electricity are more limited in number and variety of topics. They have addressed children's ability to appropriate the idea of electric current [15,16], to become aware of

connections between light bulb and battery and of polarity [17–21], to recognise what has to do with electricity and describe its operation and properties [18,22], and to understand the relationship between battery and circuit [23,24]. Recently, some studies reported the preparation of prospective primary teachers on the subject of electric circuits, in particular for what concerns the use of basic conceptual metaphors [25]. As will be discussed, this kind of theoretical background is particularly meaningful also with respect to the activity described here.

In spite of this research history on the didactics of electric circuits, there is still room to say something original on the theme. In fact, as explained above, the purpose here is to approach the subject of circuitry from a broader point of view, that is, both with regard to electricity, but also to other types of circulating matter. The aim is to emphasise the general structure that circuits of different physical nature have in common, for instance, the fact that an acquisition and distribution of energy takes place along the path, whether it is transported by a mechanical, hydraulic, or electric carrier.

Hence, the objective of the didactic intervention that was designed and tested, and is presented here, is not primarily aimed at gaining specific skills in electricity (or, at least, is not limited to it), but at using electricity, along with a mechanical and a hydraulic system, to refine an interpretative model—that of a *circuit*—which is much more general than its use in the electrical context. From this perspective, this work should therefore not only be associated with the research above—relating to the teaching and learning of electric circuits—but also with research dealing with the topic of energy in primary school [26–29]. In fact, regardless of the materials and dimensions involved, as exemplified above, what all types of circuits have in common is that there is always a distribution of energy along a path.

Moreover, emphasising the aspect of energy within the generalised subject of a circuit— and thus regardless of the materials that make up the circuit or that act as agents in it for the distribution of energy (e.g., marbles, water, air, electric charges, i.e., what is sometimes presented in the primary context as a *force of nature* [30,31])—introduces children to the theme of sustainability, in accordance with a teaching philosophy ("system thinking") that aims to provide tools that are as versatile as possible and capable of capturing the connections between different areas of knowledge and between problems. It is our basic conviction, in fact, that encounters with physics topics, especially at the primary school level (but also in higher grades), must first and foremost provide an opportunity to form and develop an efficient cognitive style, rather than provide more or less limited and approximate notions. In this case, a well-structured didactic path that is configured as a founding moment was proposed in which, through successive steps of identification of similarities and formulation of analogies, an overall principle (that of circuit) is generated that started to become a cognitive tool for the investigation of further experiences and the conferring of meaning on them. This is done by stimulating the observation of analogies between the different states in which a specific kind of circuit can be found and those between circuits of a different nature.

Therefore, the question here was whether and how it is possible to teach, or perhaps to stimulate the onset of, an analogical approach in young learners and, in particular, whether basic science education—intended for all and possibly of high quality—should and could aim to foster such a cognitive ability that can be of cultural and educational interest not only to the few destined to work in a STEM field, but for all.

The intervention consisted of an expert-led course focusing on hands-on experience with different types of circuits. Afterwards, the pupils' answers to the follow-up questionnaires were analysed. Methodologically, the research was designed as a qualitative and quantitative single-case study.

The paper is organised as follows: After summarising the theoretical framework and the pedagogical philosophy behind this study in the second section, the objectives of the research paper are provided in the third section. Section four is devoted to the description

of materials and methods. Results and their discussion are reported in sections five and six, respectively. Finally, some conclusions are provided in section seven.

## 2. Cognitive and Pedagogical Framework

### 2.1. On Abstraction in Physics

Historically, as highlighted by Koyré's milestone studies on Galileo's philosophy of nature [32], the process of abstraction that underlies the construction of a model is perhaps the most significant factor that marks the birth of modern science.

The peculiar, selective gaze cast by physicists on reality is well described by the following words of Holton and Brush:

"Consider for a moment what an amazing thing has actually happened. First, we watched the actual motion of a car along a straight road. Then, from the multitude of ever-changing impressions—the blur, the noise, the turning of wheels, the whole chaos of events progressing in time and space—we have rescued two measurable quantities, $s$ and $t$, both of which take on different values every instant, and we have found that their ratio is constant, an unchanging theme underlying the flux of otherwise meaningless, unrelated data. We have defined a concept, speed, and so have been led to discover a simple feature in an otherwise complex situation. Perhaps familiarity with the concept of speed prevents you from appreciating this experience of creating order from a chaos of sense impressions by abstracting from it some measurable data and by perceiving or inventing or intuiting a suitable concept to describe that portion of the total phenomenon [this method is] the very heart of scientific procedure, again and again" [33].

In summary, Holton and Brush conclude that "science has grown almost more from what it has learned to ignore than from what it has taken into consideration": To grasp reality in the most convenient way, scientists abstract, that is, they make reference to things that do not exist in the ordinary sense of the term, i.e., things that possess a great explanatory power but are "invisible" in the broad sense, namely, that are beyond the possibilities of our senses [34].

This cognitive process therefore emphasises imagination: to learn, it is not enough to passively receive information, but it is necessary to mobilise the cognitive resources one already possesses, and which are constantly being refined by this process of discovering.

At the same time as the important variables are extrapolated from the phenomenon under investigation so that the latter is replaced by a model that accounts for its basic structure, other different real systems and phenomena are brought back to that very same model. The gaze of physics is thus also characterised by seeing unity in diversity.

### 2.2. The Role of Image-Schemata

Let us now ask ourselves whether this major prerequisite of scientific development is at least potentially within the grasp of even the youngest children. Are pupils capable of making abstractions? Do they possess already abstractions, perhaps even very simple ones, in their cognitive background? The answer given by the cognitive sciences is positive [35–37]: The children who offer themselves to the teacher's listening are not empty vessels to be filled, but are laden with *image-schemata* developed from the very beginning of their physical experience with the world. These schemata work as a source of elementary conceptual metaphors that enable the acquisition of the first rudiments of a cultural organisation of physical reality. Categories such as SUBSTANCE, CONTENT/CONTAINER, and AGENT/PATIENT; spatial organisers such as PATH, VERTICALITY, and CENTRE/PERIPHERY; or conceptual organisers such as POLARITY (hot–cold, high–low, good–bad, etc.) are samples of elementary abstractions that children continuously project onto the world around them, testing and refining them more and more.

The existence and use of these categories allows for an important reflection on the possibility of engaging in experimental scientific discourse with children. Indeed, regarding primary school, the question arises as to whether the typical physics approach of drawing analogies, reducing, abstracting, and synthesising can be fostered by the teacher, and if so,

how can it be done and whether it can be learned. Obviously, unlike what happens in higher education, the impossibility of using mathematics—per se a tool of synthesis and hierarchy—makes the question even more difficult (and, therefore, fascinating). Nevertheless, the narrative approach typical of primary education is not necessarily a poor one in terms of scientific possibilities [31]. Even if one cannot rely on the unambiguous relationships provided by formal mathematics—incidentally, anyone with teaching experience knows that this absence of ambiguity is, in the real school context, purely ideal and, in practice, turns into its exact opposite—every-day language does possess a logical structure. In particular, it includes a whole series of universal metaphorical references, precisely the above mentioned *image-schemata*, that are, by their very nature, basic tools of synthesis. Lakoff and Johnson use the expression "imaginative rationality" to describe this peculiar situation [38]. Since children do possess basic tools of imagination coming from their bodily experience and interaction with the environment, a teachers' task is that of guiding them to use these tools to the fullest [39]. Evidence confirms that future teachers are very aware of the importance of being instructed in these aspects during their training in science teaching [25,40]. Moreover, it is precisely because these *image-schemata* exist, functioning as a kind of ubiquitous invariant of cognition, that it is possible to work out analogies and, with them, to set up abstract models for physical systems: "different phenomena are structured metaphorically using the same set of schemas. This makes phenomena that do not have anything in common objectively similar to the human mind. As a result we see them as analogous" [30].

Among these *image-schemata*, one also finds that of CIRCUIT [37]. As already described, in life as in school, there are many occasions when children are confronted with situations that are described, explicitly or implicitly, with a more or less in-depth circuit concept. In this way, such experiences are endowed with an identity and a relationship with the cognitive universe that children carry within them and that expands every day. Elementary scientific experiences should leverage these basic *image-schemata* to bring pupils to grasp parts of natural reality and, at the same time, enrich them with further attributes.

The activity we designed and whose outcomes are described here aims to facilitate this cognitive transition in pupils.

Therefore, it is on the one hand a question of illustrating the essential aspects of the electric and hydraulic circuit, and on the other of highlighting the general modelling aspects, in particular the qualitative energy balance characteristics. Thus, in the planned learning path, the asked question was if and how it is possible to make use of these structured and structuring nuclei of natural language to draw analogies across physical systems that are characterised by substantial differences but are in some ways similar to each other.

### 2.3. Analogies at Different Levels

Being able to identify and use analogies is one of those skills that makes cognition particularly efficient. The centrality of analogy as a cognitive task has been revived and reconceptualised at various times and in different scholarly contexts [41–44], including teaching and learning in students and pupils [45–49], with regard to scientific learning [50–54], as a strategy to communicate the nature of science [55] and, specifically, in the teaching of the principles of electricity [7,56–58] and also in teacher training [59]. Concerning the teaching of physics, the topic of analogy is interesting not only from the point of view of the skills that learners need to develop, and what the strategies applied by the teacher to pursue this may be, but also as a skill that teachers themselves can refine and use to make their teaching more attractive and inclusive, for example, by drawing analogies between physics concepts and themes or situations typical of other fields of knowledge [60–62].

In particular, the search for analogies is a key cognitive process that fuels scientific abstraction. Let us consider, for example, what happens when the fall of a feather and a hammer is studied. The two objects are, under the specific conditions of free fall, treated as analogous, even though they are profoundly dissimilar (e.g., it would never occur to us

to hit a nail with a feather). Just as with metaphors, also when drawing an analogy some aspects of the two objects put in relation are enhanced while others are overshadowed [38]. To Gentner and Jeziorski's definition of analogy as "a way of noticing relational commonalities independently of the objects in which those relations are embedded" [42], we add that, when an analogy between two objects is drawn, a third element is implicitly introduced, consisting precisely of the aspects common to the two objects it bridges and nothing else. This third element is necessarily abstract, i.e., it does not exist in the ordinary meaning of the term. (Note: it is abstract but is intrinsically built on categories developed by our mind, which is an embodied mind [36].) In our example above, for instance, this element is the point mass. Hence, in establishing an analogy between objects, an abstraction is used, and a *model structure* is generated.

To further emphasise the centrality and power of analogy in the growth of scientific knowledge, let us think of another striking instance in the history of physics, that is, the birth of its most central concept, energy. Its development in the mid-nineteenth century led to the foundation of physics as a unitary discipline from branches (mechanics, thermology, optics, electricity, etc.) until then independent of each other, with each branch addressing one or more specific *forces of nature* (substance, fluids, wind, heat, electricity, linear and rotational motion, gravity, etc.) [30]. Not only historically, but also every time the concept of energy arises in the mind of a new student, a powerful analogy is drawn between processes underlying phenomena of different natures. By making reference to the cognitive process known as *Figure—Ground Reversal* [63,64], one could say that what happens is that all perceptual details of the system under consideration—be it the electric wire, a spinning wheel, a moving body of any size and composition, etc.—fade into the background of our experience, while an invisible agent (the *force of nature*) and an invisible quantity (the energy it receives from, or it transfers to another *force of nature*) come to the foreground in our mind [29].

Gentner and Jeziorski [42] show how the skill to draw analogies, even among natural philosophers and scientists, has changed over the centuries, hence inferring that its peculiarities are not innate. Vendetti et al. [47] claim that the analogical ability is "critical for success in education", though it rarely develops spontaneously [65] and is difficult to acquire and use for children until the late adolescence. These researches suggest that primary school children need structured guidance when looking at areas between which to make relational comparisons.

The expected analogies to elicit in these pupils are at two different levels.

The first level stems form observing the various states in which a given system can be found when a given process occurs. In the first system we introduced to the children, a certain amount of matter (marbles, water, air) was released and allowed to flow in a tube, causing a wheel to rotate. Then, it was put back by hand on top of the tube to restart the process again. This system is characterised by input quantities—the inclination of the tube and the kind and amount of matter—and an output quantity, that is, the rotation speed of the wheel. Input quantities may be varied, and output quantities may change accordingly. Of course, to the pure phenomenological description, the system presents also many other variables that are perceived by the observer, such as the colour of the marbles; the noise they make; the weight and density of the flowing matter; the order in which the marbles proceed; the temperature of water, air, and marbles; etc. The first expected analogy from the children is simply that which occurs among all the possible pairs of input values and the corresponding output values, e.g., [high tube's slope, high wheel's speed] is analogous to [medium tube's slope, medium wheel's speed], etc. These pairs are all quantitatively different, but represent contingent instances of the same and only physical event (which, with older students, would be described through a single formula): In all cases, something comes down the tube, makes the wheel rotate and then the process can be repeated as long the initial conditions are set again, regardless of the specific values one may give to the quantities involved (and the kind of matter used). This kind of basic, banal analogy can be seen simply as a similitude or, as Hofstadter nicknamed it, a *banalogy*: one sees the same

structure in the same arrangement of matter. It might be banal, but like all true analogies, it possesses nevertheless the characteristic—typical of analogies [66]—of triggering the generation of inferences. Indeed, the observer may wonder what output (i.e., rotation speed value) will be matched to an input that has not yet been experienced, and such a question may prefigure itself as a true conjecture with exploratory value of the "matter going down-rotational speed" model that has been formed in the mind of the observer.

It is worth noticing that the above analogy comes actually into two steps. The first, more trivial, concerns the results offered by the tube as the conditions of the same kind of material flowing through it change; the second, less trivial, concerns the linking of what happens in the tube with marbles to what happens in the same tube with water or air. In this case, experience with one type of matter allows one to make inferences about what might happen with another type of matter.

This kind of (b)analogy is expected also from the observation of what happens in the second system that was brought to the attention of the pupils. This is a simple electric circuit in which one or more batteries in series allow electricity to turn on one or more light bulbs in series. Even for this system, one can vary the specific value of the input (number of batteries and lights) and observe various specific results (high or low light brightness), and realise, however, that all these states of the system are analogous to each other, i.e., they are all particular cases of a general cause and effect relationship that associates input and output in a certain way (again, something that would be described, at other levels of education, with an equation). Also for the electric circuit, the first-level analogy comes into two steps, with, again, the first step more banal than the second one (although the second one is easier than in the case of the tube): First comes the mutual assimilation of circuits composed of the same materials but in which there are quantitative variations in input values; then comes the likening between the category of macroscopic circuits and that of circuits drawn or glued on paper, which are made with different materials (as will be explained below).

Still taking into consideration the first-level analogy, and referring again to the model of cognitive development proposed by Egan [39], we may consider the observation of such processes occuring in the above described systems—in which one parameter is varied at a time and we take note of the consequences—as the generation of a sequence of samples all belonging to the very same *collection*. In the collection, all elements have a strong feature in common (a process with its own peculiar traits), while a whole set of secondary details changes: The tests carried out with the tube always consist of something flowing into it and resulting in the rotation of the wheel. Each trial is then slightly different from the other, for example, because the tube has a steeper slope or because the nature of the object that flows into it changes, just as in a collection of toy cars there are smaller or larger cars, or of different makes, colors, etc. As was said above, even in this single instance, the idea of an abstract model summarising the essential characteristics shared by all the elements of the collection cannot but arise on the child's cognitive horizon. The same holds for the tests that can be carried out by having a small electric circuit at hand: It can be made longer or shorter, with more or less light bulbs or batteries, but we are nevertheless sampling the same collection. The main *image-schemata* involved in the explanations of the various observations, i.e., CIRCUIT, FLUID SUBSTANCE, LEVEL DIFFERENCE, HIGH/LOW INTENSITY, INTERACTION, AGENT/PATIENT, help the child enormously to grasp the analogy between the different states of the system, both in the case of the tube and that of electricity, separately.

More challenging is the second-level analogy. In this case, the aim is to elicit an analogy between systems that present themselves radically different to sensory perception: on the one hand, we have the category "matter sliding in a tube", and on the other hand, we have the category "electric circuit". In this regard, it is interesting to note that the use of air as an agent in the tube can help overcome an important obstacle that could impede analogical reasoning between the two systems. In fact, using air provides an invisible current, and this could facilitate a first step towards imagining the flow of electricity, which is equally invisible in the electric wire. In short, as far as the tube is concerned, we have a circuit

in which the materials and observable characteristics are completely different from what happens in the case of the electric circuit, but the two systems harbour processes that share some important aspects with each other: an agent (marbles/water/air or the electricity) contained in a container (the tube or the electric wire) is loaded with energy (provided by the operator or the battery), which is then used to make something work (the wheel or the light bulb). When these aspects emerge, at the same time all the (enormous) differences (visible vs. invisible things; the dimensions and materials involved; the operator being a person vs. the battery being an inanimate object; something that rotates vs. something that makes light and heat) are reduced to second-order details: this is a remarkable intellectual achievement, indeed, by no means taken for granted. If, in fact, speaking in metaphors, in the first-level analogy, it was a matter of assimilating a cow to another cow, and then a cow to a horse; here, it is now a matter of putting the cow, the horse, and any other animal being into one single category. If this analogical recognition (between two types of systems that appear fundamentally different to sense analysis) takes place in children, then their *image-schemata* (CIRCUIT, FLUID SUBSTANCE, LEVEL DIFFERENCE, HIGH/LOW INTENSITY, INTERACTION, AGENT/PATIENT) acquire a higher level of abstraction and, at the same time, a greater cognitive potential, ready to be taken to a further level of abstraction and generalisation through further experiences (e.g., the water cycle, or the production of goods and the market).

### 2.4. Age and Characteristics of the Pupils

We were not interested in studying the results in correlation with specific cultural, economic, and social characteristics of the children. In any case, we involved children living in a small Italian city characterised by an integrated multicultural and multilingual reality. Instead, we took into account only one parameter that we considered fundamental, namely, age. It is precisely according to this parameter, in fact, that the pedagogue Kieran Egan distinguishes the evolution and achievement of the different cognitive planes in a person's development. The choice of our target group is by no means random: around age 8. Indeed, according to Egan [38,67], between 5 and 10 years of age is when an individual begins to detail the whys about the events provided by experience, to explore the possibilities and ranges of action, and to understand which versions of a certain phenomenon are possible and which are not. Egan refers to this age as that of *romantic understanding*: Children begin to come to terms with the limits of reality, but also with abstraction as they start to devise objective ways of referring to the world, first of all through literacy itself, that is, with the acquisition of letters, numbers and the relationships between them. Typically, this development is also made explicit in the propensity children of that age show for collecting, which basically means taking note of the possible variations of a theme and forming an abstract idea that summarises these variations and does not offer itself to perceptual experience.

In order to give more depth to the aspect concerning the development and use of analogies through age, we used a second sample of children, smaller in number (only 12) and older (middle school children, aged between 11 and 13), which we conducted in a smaller version of the course (the one more specifically aimed at the development of analogies between different kinds of circuits).

## 3. Objectives of the Intervention

As described above, the process of abstraction is based on the recognition of an analogy between different objects or events; in turn, the ability to grasp an analogy is based on the use—increasingly refined—of *image-schemata*, i.e., primitive conceptual metaphors. Our research therefore focuses on creating an activity, and evaluating its results, that stimulates pupils to use such *image-schemata* in a context of experimentation with the physical world. Summarising what has been said, the specific objectives of the structured pathway presented to the children are as follows:

- There is an objective on a general level, which is that of introducing children to the scientific process of discovery through the identification of certain characteristics of a system that can vary, causing other characteristics to vary. In this regard, the experiment with the tube (focused on the CIRCUIT aspect) lends itself very well. Indeed, in its simplicity, it is paradigmatic of an enormous class of phenomena, because, in practice, in most or perhaps all physical phenomena, there is something that possesses energy and then transfers this energy to something else.

- The second objective is to research the occurrence, degree, and extent of children's ability to identify analogies, develop them, and use them properly. The proposed research questions (RQ) were as follows: RQ_1) To what extent are children able to create the (first-level) analogy between different states of the same system, abstracting from them to produce a general model (e.g., the mechanical circuit; the hydraulic circuit; the electric circuit)? Are they able to refer to that model to formulate conjectures on possible outcomes of future experiments and aswer questions? RQ_2) To what extent are children able to create a (second-level) analogy between different systems (the mechanical/hydraulic circuit and the electric one), as well as to draw parallels between the parts of one system and those of the other system (e.g., slope plus operator vs. battery; wheel vs. light bulb; tube vs. electric wire)?

In addition to introducing basic aspects of the scientific approach (such as the above concept of circuit) and study the reactions of the pupils to them, one further purpose of the project is to familiarise children with the rudiments of electronics (especially the use of sensors, which will be central in the subsequent phase of the project), such as the electric circuit, LEDs, coin cells, and conductive ink. Furthermore, another goal is to produce a prototype suitcase containing teaching materials that can be used by primary school teachers autonomously with their classes.

## 4. Intervention Materials and Methods

### 4.1. Intervention's Details

As stated above, our major goal was to lead the pupils into finding similarities between what happens in different physical systems: A tube in which marbles, water, or air can flow; a large electric circuit (whose elements are held in hand by children forming a circle that occupies a room); or a small electric circuit (whose elements are glued or printed on a sheet of paper). In all three systems, we are dealing with concepts involving some basic metaphorical schemes that children already possess and use (albeit perhaps in an intuitive, approximate but entirely spontaneous manner), such as CIRCUIT, FLUID SUBSTANCE, LEVEL DIFFERENCE, HIGH/LOW INTENSITY, INTERACTION, AGENT/PATIENT. The three systems are closed circuits in which the effect (rotation of the wheel, switching on a light bulb) is due to the presence of something moving along a path driven by a LEVEL DIFFERENCE, i.e., the gravitational potential and the electric potential difference, respectively.

The activity was organised in three meetings, each lasting about one hour and a half. At each meeting, one of the authors, sometimes assisted by another one besides the class teacher, presented a situation and involved the children, with an increasing level of active participation from meeting one throughout meeting three: In the first encounter, children participated especially by observing, discussing, and writing; in the second, by holding the material, designing, discussing, and setting macroscopic electric circuits; in the third, by completing preprinted circuits with coin batteries and LEDs and also by manipulating them, as well as discussing. Between the first and second meetings, and between the second and third ones, children answered two questionnaires. Another data source that was used during this research was observation. In this case, certain issues were considered, such as the engagement of pupils during the activity, the answers given to some questions during the experiments, and the feedback from teachers at the end of the meeting.

The activity was conducted with six classes of children aged 7 or 8 (and a very few of age 9), for a total number of 119 pupils. The first two classes (third graders) were visited two times, as a first activity's trial and assessment, thus proposing only the first two meetings

of the project, then with the other four classes (three third grade, one second grade) was the whole three-meeting project proposed. Each visit lasted approximately one hour and a half. The activity took place during the COVID-19 pandemic period and therefore both children and adults wore face masks.

Moreover, a shortened, one-meeting-only version of the course was proposed to a group of 12 middle school children, aged 11 to 13.

In all the meetings, pupils were stimulated to get involved and invited to ask and answer questions.

In order not to interfere too much with the class teachers' annual planning, we left it up to them to decide whether to have the two questionnaires filled in at home or in class after the visit. In any case, it was recommended that children carry out their task autonomously and independently from one another. This sets obviously a possible intrinsic limitation to the validation of the results. However, we found that the pupils who completed the questionnaires at home achieved statistically the same results as classes who completed the questionnaires at school in the presence of teachers instead.

In more detail, the three meetings were organised and conducted as described below:

- First meeting: Introductory talk citing and explaining words such as observation, measurement, movement, science (what does a scientist do?), making examples, and urging the children to refer to their own experience; description of the apparatus (the tubes); experiment with marbles varying the tube's slope and then varying the quantity of marbles and writing the results in the form of a table at the chalkboard; before each attempt, pupils were invited to express a conjecture about what was going to happen; experiment with the second tube and water, varying the slope and then varying the quantity; experiment with air (not for all classes), by varying the pressure of air at one end of the tube via an inflated balloon. Questionnaire 1 (see Section 4.2) asked the pupil to draw the apparatus by naming its parts and explaining what was happening in it (with marbles or water), and to answer a question.

- Second meeting (a week later): A choral recapitulation of what was seen in the previous meeting; a dialogue on electric current and its uses in the pupils' experience; simple explanation of the modules; in small groups (4–5 groups per class, each group with 4–5 children) circuit construction: each group is asked to form a circle by connecting the various elements held in their hands, firstly only with one battery and one light plus the conductors, then one battery and two lights (in series), two batteries (in series) and one light, two batteries (in series) and two lights (in series); the children were asked to observe any differences in bulb brightness obtained in the different configurations and take note of them. A qualitative table of these outcomes was written on the chalkboard. In the dialogue about what happens in the circuit when the number of batteries and lights varies, the instructor explicitly proposed a reminder of what happens in the tube when the inclination varies. The instructor explicitly drew parallels between what happens in the two different systems and between the quantities involved. Children were then asked to observe what happened when the circuit was not closed or was closed with certain materials such as pieces of wood or plastic pens or metal scissors. On the subject of electric conduction ("something that flows in the wire like marbles or water in the tube, but you cannot see it, like the air that was imagined flows in the tube"), many materials already available in the classroom were tested by using the "funny tester" and/or placing them between two modules of the circuit; finally, one or both (depending on time left) electric board games were shown and played a little with, also to make pupils consider that the abstract things they were experiencing can be applied. In Questionnaire 2 (see Section 4.2), they were asked to draw a circuit and try to explain what happens, plus a series of exercises described above.

- Third meeting (after another week): Summary of the observations made in both previous meetings; description of the materials to be used (wires, coin cells, LEDs, conductive ink, and tape), and explanation of the tasks to be carried out in small groups (two or three pupils), i.e., the construction of four electric circuits like the

macroscopic ones kinaesthetically experienced the meeting before—this time with LEDs and coin cells glued on paper—i.e., changing the number of lights and batteries (both in series). Finally, a conversation took place about what was seen in the whole path, particularly going back to what was asked in the second questionnaire and offering a confirming view on the analogy between the systems seen and a parallel between the roles played between their parts and between the processes that happen in them.

The activities with the class of middle school children were conducted in a similar way but concentrated in a single two-hour meeting. With these children (aged between 11 and 13, with an average age of 11.7 years), the tube with marbles and water was used—only narratively, without proposing experiences—and the analogy with electric circuits, which they knew from a previous activity. They were then provided with Questionnaire 2, in a slightly modified version, but one that basically followed the one described below.

*4.2. Materials*

Commercially available materials and self-designed and manufactured elements were used. In the first meeting with the children, the observation of phenomena concerning a transparent tube in which glass marbles can slide (Figure 1) was proposed. The tube had a variable inclination. The inclination was able to be measured by the height point from which the marbles are released. This height was marked by the number of wooden blocks below (h1 to h4). Along their run, the marbles hit the blades of a wheel whose rotation speed was measured by an IR speed sensor located behind the wheel and displayed both as a number (RPM) shown on an LCD and as a light vertical bar consisting of 4 LEDs for younger children unable to read numbers, with each color corresponding to a preset range of speeds: 0–50 yellow, 50–100 green, etc. The data acquisition was performed using an Arduino UNO board powered by a commercial powerbank. The same kind of observation was also designed in such a way to use, instead of marbles, water or air (by placing an inflated balloon at one end of the tube). The spinning wheels were designed and 3D printed specifically for this project. Two different spinning wheels and tubes were needed for marbles and water, whereas the experiment with air ran well with both.

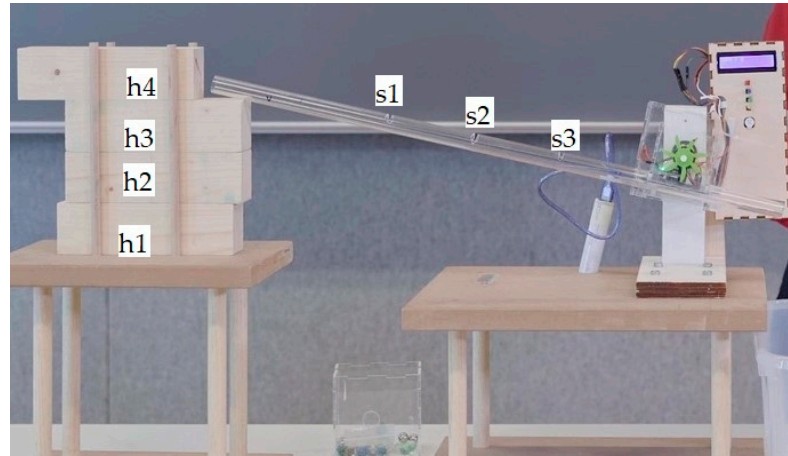

**Figure 1.** The tube for marbles. The marbles are inserted on the left, where the end can be rested at different heights, identified by the wooden blocks (h1 to h4). Available starting points are indicated with s1, s2, and s3. On the right is the wheel to which the speed sensor is attached. For water, the apparatus is similar, but with a container and a tap at the left end.

For the second meeting, wooden modules were built that pupils had to hold in their hands while forming a circle, so that both the number of bulbs (in series) and the number of batteries (in series) could be easily changed (Figure 2). Each module consisted of a stick with ends wrapped in aluminium foil and an electric wire connecting these ends with the

element taped in the middle of the stick. In particular, there were six available battery sticks (that is, a wooden stick with a 4.5 V rectangular battery taped in the middle), six light sticks (i.e., with a small light bulb taped in the middle), and twenty pairs of conductors handles (that is, two shorter wooden sticks connected to each other by a 40/50 cm long electric wire and with ends wrapped in aluminium foil). Besides these modules, a "funny tester" was prepared made from a beaker of yogurt from which a "nose" consisting of a small light bulb may light up the moment the "hands" of the beaker close the circuit (powered by a battery inside the beaker) and two follow-the-path board games. In these games, when the player makes a wrong move, a circuit is closed lighting up a light bulb. Finally, a circuit model in which one or two batteries (in series) and one or two light bulbs (in series or in parallel) are fixed on a square wooden board of about 30 cm side.

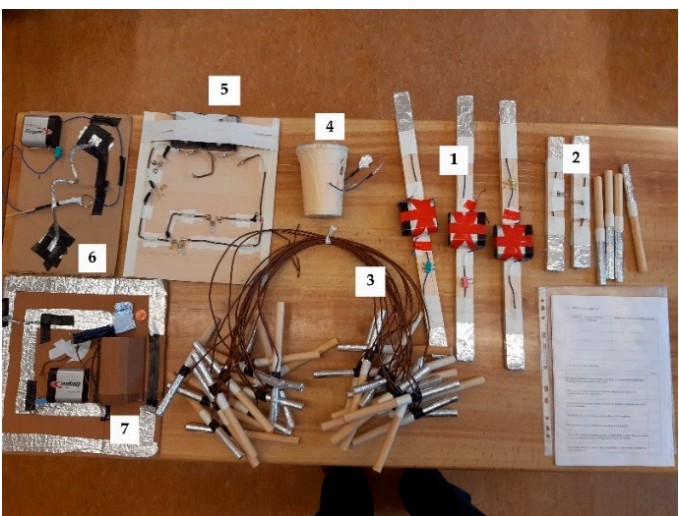

**Figure 2.** Materials for the second meeting: 1. battery sticks; 2. light sticks; 3. conductors handles; 4. funny conduction tester; 5. circuit model; 6. 3-D follow-the-path board game; 7. 2-D follow-the-path boardgame.

In the third meeting, pupils had at their disposal electric wire, pens with conductive ink, conductive tape, LEDs of various colours, and coin cells. There were also sheets available in which some sections of the circuits were already screen printed using conductive ink, i.e., silver ink.

The two questionnaires consisted of a series of open-ended questions and small assignments. Written questionnaires, rather than interviews, were used, in order to give the children the opportunity to reflect calmly on their answers, at their own pace, and possibly to correct themselves, without them being prompted to do so (as they might in an interview, where there is also non-verbal communication that has a certain effectiveness). In addition, we proposed questions and exercises of different types (completing sentences, linking sentences, multiple-choice or true/false type questions, and drawing and narrating in one's own words), so as to diversify the analysis and reduce the possibility of some form of bias (e.g., children used to a certain type of exercise but not another).

To answer the questions, the pupil must activate the basic conceptual metaphors they possess, adapting them to the new context and using them to link different contexts together.

For example, if Sentence 2.4 (see below) is correctly completed, it indicates that the pupil has used the AGENT/PATIENT *image-schema* correctly: he/she identified that the inclination of the pipe is the AGENT, and that the current (of marbles, of water) is the PATIENT. But not only this, it also indicates that the pupil also used the same *image-schema* AGENT/PATIENT in the case of the electric circuit. The use of the same *image-schemata* in the two different systems (the pipe, the electric circuit) implies that the pupil may eventually have drawn an analogy between these two systems, and this is checked through the answer to Sentence 2.11 (see below).

The description of the questionnaires in detail follows:

Questionnaire 1 asked for drawing the apparatus and explaining what was done with it:

**Question 1.1:** *Sketch the apparatus and indicate its parts (tube, wheel, led lights, . . .) with arrows.*

**Question 1.2:** *Describe the experiment we did with marbles and/or water.*

**Question 1.3:** *How do you make the wheel spin at maximum speed?*

Questionnaire 2 was longer and more detailed. As for Questionnaire 1, the first part was about drawing the apparatus and explaining what was done with it:

**Question 2.1:** *Sketch an electric circuit and name its parts.*

**Question 2.2:** *How would you explain to a friend of yours how the circuit works?*

The second part of Questionnaire 2 comprised fill-the-gaps sentences with words to be chosen among the given ones (the pupils had to choose 6 words out of 8 and place them in the correct spot):

**Sentence 2.1:** *The more the tube is inclined, the . . . turns the wheel.*

**Sentence 2.2:** *To turn on the light bulb, you have to . . . the battery to the circuit.*

**Sentence 2.3:** *. . . and water only go down if the tube is . . .; the bulb will only light up if the circuit is closed and if it includes . . ..*

**Sentence 2.4:** *Inserting the battery into a circuit to make . . . flow is like raising one end of the pipe to make . . . come down.*

Given words: *inclined–slope–marbles–current–faster–connect–battery–electricity–water.* (Notice that the exercise was a little more difficult that the usual because in such kind of exercises normally all the words given are supposed to be used.)

The third part of Questionnaire 2 consisted of "true or false" sentences:

**Sentence 2.5:** *If the tube with the marbles is at a steep angle, the marbles descend very quickly and the wheel spins very fast.*

**Sentence 2.6:** *If I have a lot of batteries in the circuit, the electric current is more intense, and the bulb produces a very strong light.*

**Sentence 2.7:** *If you don't push them, the marbles can only spin the wheel if the tube is on an incline.*

**Sentence 2.8:** *If the tube is horizontal, the water cannot flow, even if it is a lot.*

**Sentence 2.9:** *In the circuit, electric current only circulates if there is a light bulb.*

**Sentence 2.10:** *Electric current circulates in the circuit only if the battery is there.*

**Sentence 2.11:** *Electric current is like the current of marbles; the electric wire is like the pipe; the battery is like the slope of the pipe; the light bulb is like the spinning wheel.*

**Sentence 2.12:** *The electric current is like the pipe; the light bulb is like the marbles; the battery is like the little wheel that turns.*

The fourth part of Questionnaire 2 required children to connect with lines "*situations that seem similar to you*": to each situation included on the left column a situation on the right column (Table 1).

**Table 1.** Fourth part of Questionnaire 2. It focuses on drawing parallels between the two different systems.

| | |
|---|---|
| *Spinning wheel in the tube* | *Battery in the circuit* |
| *Inclination of the tube* | *Bulb that lights up in the circuit* |
| *Tube with low inclination* | *Movement of marbles or water in the tube* |
| *Circuit with two or more batteries* | *Tube with high inclination* |
| *Electric current in the circuit with the battery* | *Circuit with a single battery* |

## 5. Results

The qualitative assessment of the children's level of participation was limited in its non-verbal part by the face masks. It was possible to observe the level of attention through body language, but certainly not smiles or grimaces of displeasure. This said, from observing the children during the meetings, it was possible to conclude that there was a strong interest, attention, and participation in the activity. Conversation was relentless, and replies to inquiries numerous, as well as questions, remarks, feedback, and even proposals of new trials. For example, just as a result of one of these exploratory talks, at the urging of the children, the possibility of having the marbles start from different points on the tube was added to the tube with marbles (s1, s2, and s3 in Figure 1). This was something not initially included among the interesting variables with which to experiment.

Excitement was generated even by simple events, such as guessing the outcome of a trial or succeeding in turning on a light bulb after managing to get all the modules to touch (Figure 3), or after seeing the ink actually become conductive.

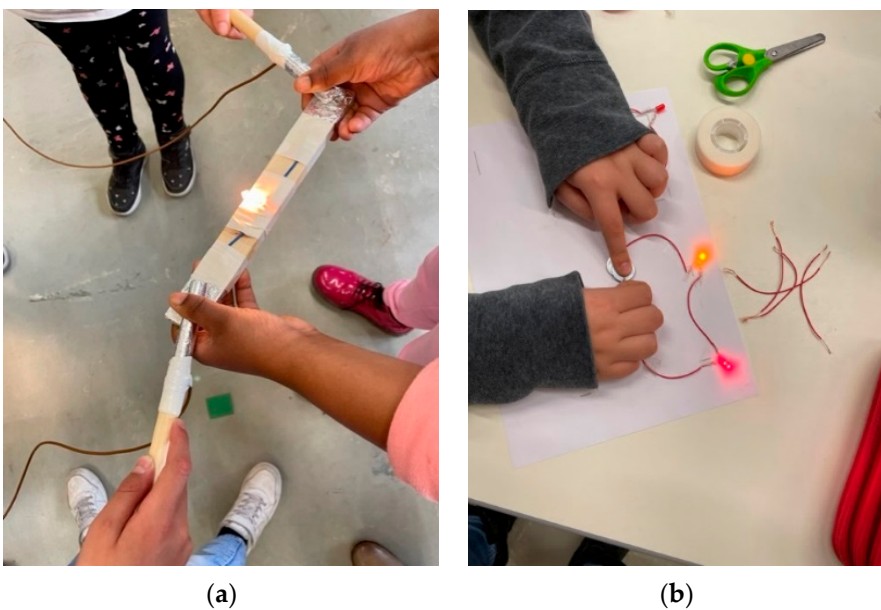

| (**a**) | (**b**) |
|---|---|

**Figure 3.** (**a**) Detail of children making contacts between the elements of a circuit. (**b**) A circuit with two coin cells, electric wire, printed conductive ink, and LEDs.

It must also be said that an hour and a half of activity proved to be just the right amount of time, because towards the end of the interventions, the classes began to show tiredness and the first signs of distraction (such as, for example, resting the head on the table, seeking the complicity of classmates to chat or play, looking out the window). In this respect, a first clear difference between the second graders was seen, who were more prone to getting tired earlier, and the third graders.

Teachers expressed much appreciation for the activity. We collected clear evidence that they collaborated very actively and competently (for example, offering to repeat things said, or trying to motivate less proactive children), especially during the more demanding

meetings (second and third ones). Therefore, concerning the purpose of developing a prototype suitcase with materials for teachers to conduct this experience, on the basis of our experience, we can say that they would certainly be able to run it well on their own with their classes.

As described above, each weekly intervention began with a dialogue-based introduction or a summary of what had been seen the previous time. By answering the questions, correctly summarising what had been seen, intervening in the answers of others by adding further details, the pupils have shown that they remembered well the previous experience(s), sometimes even in the aspects on which little time had been spent. This certainly helped to give continuity and effectiveness to our intervention.

Again, limiting ourselves to what has been observed in the dialogues, we can definitely argue that the children were able to generate inferences from observing the proposed attempts, both in the case of the tube (with the different materials) and in the case of the circuit. Correct or incorrect, the emergence of such inferences is, as mentioned, symptomatic of the production of an analogical process.

Let us now turn to the analysis of the responses to the questionnaires. Leaving aside the first two classes that served as a sort of calibration of the project, we collected 48 Questionnaire 1 and 48 Questionnaire 2 replies of the 60 of each kind we distributed in the remaining three third-grade classes. Indeed, we decided to omit from the analysis the questionnaires completed by the children of the second-grade class because we realised that for most of them, they constituted a considerable challenge just in reading the questions and actually writing down the answers. The 96 questionnaires received by the three third-grade classes taken into consideration were all completed in full. The 20 missing questionnaires (for each kind) were due to a few children who simply did not return them and to other children who were absent on the day of return or on the day of the experiment.

For both the first and second questionnaires, the first task involved drawing and freely describing the proposed activity. In both questionnaires, we included the request to make drawings because we thought they could be a way through which children could more easily appropriate what they had seen, before answering the subsequent questions, which are the actual object of our attention. In both cases, one notices a wide variety of results and levels of abilities. Two drawings, that sample the upper part of this range (the parts of the apparatus are distinguished, there are details, the proportions between the parts are respected) are shown in Figure 4.

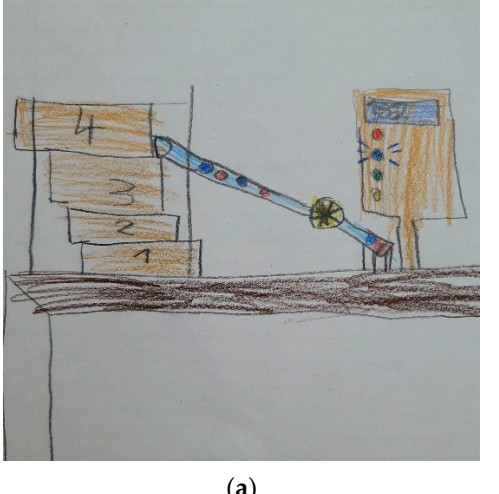
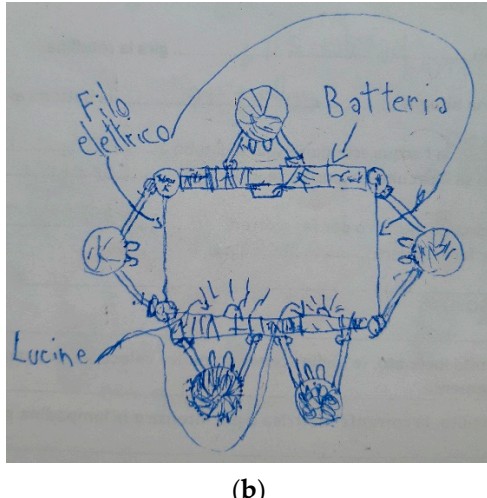

(**a**)                                        (**b**)

**Figure 4.** Two drawings by third-grade children: (**a**) The experiment with the tube and marbles. (**b**) A macroscopic circuit with electric elements held in hands by pupils (as seen from above).

As for the first questionnaire, after the request for a drawing and explanation of the experiment, the attention was focused particularly on the third question, which aimed to

find out whether pupils were able to extrapolate, on the basis of the experiment conducted in the classroom (i.e., the configurations with input/output as actually implemented and observed), the outcome of a limit case (i.e., an imaginary item that in principle belongs to the same collection of configurations). A summary of the responses is provided in Table 2.

**Table 2.** Answers to question 1.3 "How do you make the wheel spin at maximum speed?"

| | |
|---|---|
| Children indicating the variation in quantity (number of marbles, amount of water) | 56% |
| Children indicating the change in slope or pressure | 71% |
| Children indicating both factors | 38% |

Table 3 summarises what can be deduced from reading the sentences to be completed in the second questionnaire. Notice that there was no question made on the experiment with the tube using air. This was due to the lack of time, i.e., sometimes it was not possible to have the opportunity of showing air passing through the tube, and other times it was shown without performing a complete experiment but just in a very qualitative way (the highly inflated balloon producing high speed of the wheel; the underinflated low speed).

**Table 3.** Correct understanding of the individual systems, according to the answer to the "fill-the-gaps" Sentences 2.1 and 2.2.

| Tube (Marbles/Water) | Electric Circuit |
|---|---|
| 75% | 79% |

Table 4 contains other results concerning the two systems individually considered, but valued through a different approach, that is, the true-or-false questions.

**Table 4.** Correct understanding of the individual systems, according to the answers to the true-or-false Sentences 2.5–10.

| Sentence | Tube (Marbles/Water) | Sentence | Electric Circuit |
|---|---|---|---|
| **2.5** | 96% | **2.6** | 85% |
| **2.7** | 40% | **2.8** | 81% |
| **2.9** | 69% | **2.10** | 65% |

Finally, results in Tables 5 and 6 provide us with the data by which to assess the children's ability to conduct the (second level) analogy between the two types of systems, i.e., the current of matter (solid, fluid or air) and the current of electricity, or, more precisely, between the mechanical and electric circuits.

**Table 5.** Correct understanding of the analogies between the two systems. Sentences 2.3 and 2.4 are "fill-the-gaps", and Sentences 2.11 and 2.12 are "true or false".

| | Correct |
|---|---|
| **Sentence 2.3** | 56% |
| **Sentence 2.4** | 54% |
| **Sentence 2.11** | 73% |
| **Sentence 2.12** | 73% |
| **All the four sentences correct** | 27% |

**Table 6.** Correct understanding of the analogies between the two systems: connecting the "similar" situations listed in Table 1.

| Children Who Made ... | % |
|---|---|
| ... at least 3 correct connections | 38% |
| all 5 correct connections | 23% |

Finally, it is worth noticing that three children (i.e., 6%) did all the tasks concerning the second-level analogy correctly.

## 6. Discussion

From the qualitative observation of the children's attitude, from the way they quickly learned to fill in an ordered table, we can certainly conclude that the overall objective of setting up an experiment that would introduce the children to observation and measurement, leading them to correlate two measures (an input variable and an output variable) was certainly successful. Equally positive is the impression that the class teacher can independently and competently guide this type of investigation. Hence, the tilting tube with the wheel can certainly be an excellent tool to introduce children in a simple and direct way to the systematic study of a regularity of nature.

As explained in section two, the results were analysed by classifying them according to whether they refer to the individual systems, i.e., the tube with the flow of marbles and water or the electric circuit (first-level analogy), or whether they refer to a parallelism between these two systems (second-level analogy).

During the activity, a percentage of children higher than 50% showed that they had grasped the fundamental analogies between the systems, especially for what concerns the role of the battery and the inclination of the tube, and that of the user, i.e., the spinning wheel and the light bulb (percentage higher than 70%). It is also evident, regarding the electric circuit, a fair ability to distinguish the current, qualitatively measured by the degree of brightness emitted by the bulb, from the voltage, i.e., the force provided by the battery.

To get more into detail, concerning RQ_1, Tables 2 and 3 show that a majority of the children (from 56% to 79%) demonstrated a sound understanding of what was happening in the single system (the tube circuit or the electric circuit) as the variables took different values. In particular, they understood what the role of the tube inclination was and how this affected the speed of the wheel. They similarly understood what was happening in the electric circuit, understanding for example that more batteries would have a greater effect, i.e., a greater intensity of light.

These results show that, to a large extent, the pupils grasped the presence of a cause–effect principle in each of the two systems; what the causes and effects were in each system; and that according to the values of the cause, the values of the effect changed. Above all, they appeared to be able to imagine the system in general, i.e., even without specifying one of its directly experienced configurations (e.g., specifying how many marbles and what slope, or how many batteries and how many lights). This means (and this is far from trivial, although one soon gets used to it) that the children have employed an abstract category—again, in this case, for each of the two systems taken separately—to which all possible states of the system ideally converge, seen as essentially analogous to each other. In their imagination, the situations described in the sentences to be analysed do not only correspond to the four slopes actually experienced in the specific cases of four or eight marbles but are some of the many possible particular cases in which an entirely general model can be found embodied, i.e., beyond the contingent pair of values that cause and effect have actually assumed during the classroom demonstration. Hence, it can be said that, for each one of the two systems, these children have grasped an analogy that runs among all the possible configurations that the system could be found embodied in, generalising the process that occurs in each of them. As proof of this, let us add that in conducting the experiment, many pupils suggested that the tube could also have been

tilted even further than it was, or that instead of marbles or water, other things could be used, such as rice, small sweets, or other small objects that only need to be raised to a certain height and brought to the tube inlet and somehow be able to slide down.

As might be expected for RQ_2, the analogy between the two systems—a second-level analogy—was more difficult for the children to grasp. As Table 5 shows, only about one out of four children provided sufficiently convincing evidence that they had assimilated one system to the other, thus elaborating the two into a single category, that of a generalised circuit, with all *image-schemata* involved. This, it should be noted, despite the fact that this category is supported by *image-schemata*, such as CIRCUIT, LEVEL DIFFERENCE, SUBSTANCE, INTERACTION, AGENT / PATIENT, HIGH / LOW INTENSITY, that children should already possess among their cognitive tools. More children partially grasped the analogy, understanding for example that the battery in the circuit was like the slope in the pipe, or that electricity flows like water or marbles, but then contradicted themselves by not recognising other aspects of this analogy, or vice versa. How can this be explained? Is it a failure to apply the whole set of *image-schemata* that support the creation of the model-generating analogy? It should be remembered here that *image-schemata* are developed through bodily experience from an early age but continue to do so throughout a lifetime. The ability to know how to apply them in specific cases also develops with age. Let us recall Egan's provocative viewpoint: contrary to popular misconception, children are not concrete thinkers who over time learn to become abstract thinkers, but are already abstract thinkers who learn to use their cognitive tools, as they develop, in more and more practical situations [39]. For example, the CIRCUIT *image-schema* can certainly be very elementary in children, and incorporate only a few primitive aspects, such as that of a starting point that coincides with the finishing point after a path has been travelled. By instance, the idea that the substance moving in a circuit acquires/loses a capacity precisely in moving along the path may be missing. In our opinion, this confirms our view that the teachers' task is to be aware of the primitive concepts their pupils already possess, rather than to provide them with notions, and therefore to support the children in the process of deepening and refining them through the supply of ever new experiences.

One possible objection might concern the limitations of the proposed model, in particular the "danger" of instilling the wrong model of electric current in children. In this activity, children in fact equate charges with balls or liquid flow, which starts at one point and arrives to another. To the eyes of a physicist, this naturally appears to be a gross oversimplification: there is something missing to represent resistance, there is no continuity of flow (e.g., the tube should always be full of water or marbles); to be even more precise, there is no crystal lattice, with gaps and defects, and of course electrons have many properties that marbles and water have not, and vice versa. Size is also completely misleading: an electron is many orders of magnitude smaller than a marble. Let us reply that a model is obviously something limited, something that cannot and should not reproduce all the characteristics of the real system to which it refers. If it did, it would be useless! Furthermore, the model must be commensurate with the objective (and the age and background of the students): providing too much information would lead to the acquisition of none. One must have the courage, in the education of children and students, to simplify to the right degree, and this, of course, in the eyes of the professional physicist, necessarily implies some lack of precision and correctness of the description. It will be the task of subsequent degrees of education to add detail and correct this model. On the other hand, even the so called scientifically accepted interpretation models are always open to modification and improvement. We were primarily interested here in emphasising the general idea of circuit, and in particular the fact that along the circuit there is a point at which the charge gains energy and then one or more other points at which it gives it up, and that this constitutes an interpretative scheme for apparently different situations.

Table 6 shows a result (23%) quite consistent with that of Table 5 (27%). The fact that both these two tables show reasonably close percentages could be, in our opinion, an interesting index of the robustness of the results obtained. While summarising exercises

that are independent of each other in content and methods, both indeed illustrated results relating to knowing how to connect systems of different natures, i.e., they reflect the pupils' ability to draw the second-level analogy. It should be noted, however, that only 6% of the pupils did both the tasks of Tables 5 and 6 correctly, so there were about 20% of pupils who showed that they perfectly grasped the analogy between the two systems in one type of exercise but not in the other. This confirms what was said above, that the ability to grasp the analogy is there, but it is fragile and needs encouragement and training. It is interesting in this regard to note that in the exercise of Tables 1 and 6 (the connection between similar situations), frequently pupils did not simply miss the expected answer, but made connections that, instead of highlighting the analogy between the *two* systems (the second-level analogy)—as was our expectation—highlighted the (first-level) analogy between situations involving the *same* system (e.g., connecting "circuit with a single battery" with "circuit with two or more batteries", and "tube with low inclination" with "tube with high inclination", rather than "circuit with a single battery" with "tube with low inclination", and "tube with high inclination" with "circuit with two or more batteries").

To complete and further frame this study, let us consider the results of a survey conducted on older children (a group of 12 middle school individuals, with an average age of around 12 years). After having watched the experiment with the tube—presented to them as an integrative activity to an educational path on the use of sensors—these children answered the same questions posed to the younger children (actually, with less time available and with a few more difficulties, e.g., words to complete sentences were not made available). In each type of question, they performed equal to or better than the younger children, and in general, we can firmly state that there is evidence of a significant trend of improvement in the abilities of drawing analogies (confirming Vendetti et al. [47]). For example, 80% of them completed the two fill-the-gaps questions (Sentences 2.3 and 2.4 of Table 5) correctly, compared to 55% of the third-grade pupils; 20% of them did all the tasks correct, with respect to a 6% of the younger pupils.

Regarding the children of the only second-grade class involved, the results were not analysed in depth, although this was the original intention. The participation and enthusiasm were quite similar to those of the third-grade children, but their ability to fill in the questionnaires was overestimated, which proved to be too long for children who were on average still rather slow in reading and writing. In any case, from the conversational exchanges conducted during the experiments, it was possible to observe that the children generally grasped what was going on and were able to formulate conjectures as to the possible outcomes of a test, but in a much more qualitative manner than their older colleagues (e.g., without being able to consciously formulate an order of magnitude for the speed of the wheel that would be obtained).

## 7. Conclusions

A path was developed which is centred on children's awareness of structuring analogies between observations concerning different systems. The activity was intended as a first opportunity to approach the typical way of proceeding of physics, namely, that of extrapolating from a complex multiplicity of phenomena a common structure capable of conferring a unitary and effective meaning. For this, a very simple but also very fundamental experiment was used, because it outlines the basic idea of a physical process (there is something that moves and sets something else in motion) and its prerequisites (the idea of a cause–effect principle and that of contiguity between cause and effect). Moreover, the integration of such an experiment into a circuit also allows for the idea of energy conservation—equally basic in the structure of physics—to be conveyed more or less explicitly.

The interest of this experimentation lies first and foremost in the fact that it aims not so much at the acquisition of specific content concerning nature (things such as the force of gravity that makes water fall, or the concept of conductor or insulator) but rather at the development of the very broad and general concept of CIRCUIT. This was done through the fostering of an analogical way of thinking, based on (1) extrapolating the

truly salient characteristics of a system and (2) through these characteristics comparing different systems, identifying similar things and dissimilar things. This is made possible by bringing out those *image-schemata* that are already present in the language (and thus in the cognitive system) of children. The development and stimulation of this type of thinking is fundamental in order to actively and hierarchically set up the assimilation of the contents of science that will be offered to them in the subsequent school years.

This path was experimented with six elementary school classes, obtaining encouraging results. Children, at an age when they are beginning to explore the various possibilities in which a given system—any kind of system—can present itself, if appropriately guided, know how to classify simple experimental data. On the basis of them, they start to elaborate an abstract model of the system in which physically important quantities are selected and correlated, and from which they are able to formulate inferences about further experimental outcomes. Less frequent—about a quarter of the children—is the ability to draw analogies between different systems, i.e., systems made up of different materials and in which different *forces of nature* operate but nevertheless characterised by a similar architecture and functional correspondences between different elements.

The fact that the questionnaires were very difficult to complete in some parts for second-grade children is certainly the greatest limitation of our study, both because it would have been useful to measure the results for this sample as well, perhaps in order to compare them with those of older children, and because it somehow suggests that the questionnaire may have posed basic difficulties (reading, interpretation, comprehension of the problem) also for at least some of the latter.

Finally, this research opens up interesting prospects for further investigation, e.g., designing a pathway that further expands the potential of the circuit concept in primary school and subsequent grades (not only in the physical field but also, as mentioned, towards natural phenomena such as the water cycle or scientific–economic–cultural phenomena such as energy supply and consumption); studying the development of analogical skills on this and other basic topics across different school grades; designing similar pathways centred on other basic concepts; and observing the cognitive effects of such an approach across years or school grades.

**Author Contributions:** Conceptualization, L.C., S.K., P.L. and F.C.; Methodology, L.C., S.K., P.L. and F.C.; Software, S.K.; Validation, F.C.; formal analysis, L.C., S.K. and F.C.; Investigation, L.C., S.K. and F.C.; Resources, L.C., S.K. and F.C.; Data curation, L.C. and S.K.; Writing–original draft, L.C. and F.C; Writing–review & editing, L.C., S.K. and F.C.; Supervision, P.L. and F.C.; Project administration, P.L. and F.C.; Funding acquisition, P.L. and F.C.; All authors have read and agreed to the published version of the manuscript.

**Funding:** This research activity was developed within the Free University of Bozen-Bolzano project "Discovering complexity: Advanced technologies for an education in storytelling and systemic thinking.". This research received no external funding.

**Institutional Review Board Statement:** The study was conducted in accordance with the Declaration of Helsinki and approved by the Ethics Committee of the Free University of Bozen-Bolzano (protocol code ATNEST_Cod_2022_05 dd 21.03.2022) for studies involving humans.

**Informed Consent Statement:** Written informed consent was obtained from all subjects involved in the study.

**Data Availability Statement:** The data presented in this study are available on request from the corresponding author.

**Acknowledgments:** We acknowledge the precious help given by the classroom teachers Corradina Bonaccio, Nives Fois, Irene D'Agostino. We also acknowledge the staff at the BITZ unibz fablab for their availability, support, and assistance in developing the mechanical setup used in this work.

**Conflicts of Interest:** The funders had no role in the design of the study; in the collection, analyses, or interpretation of data; in the writing of the manuscript; or in the decision to publish the results.

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
