# Peer review of "Teaching and Investigating on Modelling through Analogy in Primary School"

_education, doi:10.3390/educsci13090872_

Round 1

Reviewer 1 Report

The authors indicate their research aimed to see if approaches to complexity in physics could be taught to primary school students. These approaches included identifying relevant variables in a process, recognizing analogs of those processes (making analogies), and forming mental models for a system or class of systems. The main contribution of this work was the description of the curricular activities the authors designed and implemented, which included clear illustrations and was explained through three frameworks (abstraction in physics, image schemata, and analogies).

The overall organization and style of writing, especially within sections, made it difficult to understand how the authors conducted their research. Below, I address my main concerns with each section.

§  In the introduction, the authors investigate how “the fundamentals of the physics approach” (line 34) can be taught to young children, but do not define this expression (e.g., what counts as “fundamentals” and what is meant by “approach”). They state the focus would be on identifying variables in processes, recognizing similarities between different situations, and creating mental models of systems, which seem to correspond to three well-researched constructs (causal reasoning, analogical reasoning, and scientific modeling) but were not addressed explicitly in the review of the literature. The authors also introduced the ideas of “complexity” (lines 64-75), “energy,” and “sustainability” (lines 104-120). The lack of coherence between these terms makes it difficult to ascertain what the authors are specifically investigating.

§  In the section titled “Cognitive and pedagogical frameworks and objectives,” the ideas of “abstraction in physics,” “image-schemata,” and analogy levels were also introduced without clear explanations about how this relates to the foci. The items listed for the categories they identified for image-schemata are not necessarily mutually exclusive (e.g., centre/periphery was listed as spatial organization, but could also be considered as a conceptual organizer); explanations about how the items were sorted into each category were needed. The sub-section on “different levels of analogies” was not consistent with the topic level of the other two sections (abstraction in physics, image-schemata); a general discussion of analogies would have been more consistent, followed by their discussion of the different levels. Additionally, citations are needed for the statement about how analogies are drawn between different phenomenological processes when “the concept of energy arises in the mind of a new student” (lines 255-257). The writing on the different levels (lines 272-326) was unbalanced – with only about 20 lines addressing the second level and the rest addressing the first level. It was also unclear why the “age of the pupils” and objectives were included as sub-sections. The age of pupils seems like it would fit better in the methods section and the objectives should be a separate section.

§  In the section describing the intervention and methods, some paragraphs were redundant (e.g., information in “project path at a glance” is repeated in “intervention’s details”). One suggestion might be to re-order the paragraphs – such as starting with a description of the intervention (which could also address the materials section). More importantly, the research methodology presented was not rigorous. The research questions, presented in the objectives section, indicated a focus on investigating students’ ability to create first- and second-level analogies; however, the assessment instruments used (questionnaires) did not ask students to create analogies but to recognize them. The authors also said questionnaires were used instead of interviews so that students could have more time to reflect on their written responses, but students may not articulate everything they are thinking in their written responses. Next, some aggregated demographic information about the participants (e.g., gender, ethnicity, socioeconomic status) and the context of the study (e.g., type of school [e.g., public, private, charter]) were not presented. The items of the questionnaire were shared but did not explain which items corresponded to the abstraction of physics, image-schemata categories, or levels of analogies that were introduced earlier in the framework section.

§  In the results section, claims made about their data and analysis were not explicitly supported. Some examples included: children showing “strong interest, attention and participation in the activity” (line 614), “Excitement was generated” (line 620), and teachers “much appreciated” the activities (line 631). It was not clear how these claims were measured or what part of the surveys (the only data source mentioned) provided evidence for these claims. In another example, an explanation was lacking: the authors said a time of “an hour and a half of the activity proved to be just the right amount of time” (line 626); this claim raises questions such as why would 100 minutes be better/worse than 90 minutes? The authors shared images of students’ drawings in Figure 4 but did not present an analysis of those drawings other than to state that the drawings represent the upper range of drawings produced by students; explaining why those drawings were in the upper range would have been helpful, in addition to providing an example (and explanation) of a drawing from the lower range. The authors also stated that correct or incorrect, the students’ generation of inferences was evidence of “production of the analogical process” (line 644); it was not clear how generating incorrect responses was definitive evidence of student engagement with making analogies. The results presented in Tables 2-6 were too brief, showing only the percentage of correct/incorrect responses; it would have been helpful to present the correct responses and provide examples of incorrect responses for at least some of the items in those tables.

§  In the discussion, the authors start by stating that the objective of “setting up an experiment that would introduce the children to observation and measurement…was certainly successful” and that the “teacher can independently and competently guide this type of investigation.” An explanation for what counts as “successful” or “competent” would have been helpful. Overall, clarity in language was missing. For instance, the authors later talk about a “good percentage” and “good majority” of children demonstrated understanding. Since data was collected, it would have been more useful to state the exact percentages; additionally, an explanation of what was observed that demonstrated “understanding” would also have been helpful for readers (e.g., “understood what the role of the tube inclination was” is an inference … since the collected data were written responses that partly involved fill-in the blanks from a word bank and matching phrases from two columns, it is possible students could have guessed correctly). The paragraphs starting from line 745 were particularly difficult to follow, mainly due to the lack of organization of ideas presented in the arguments. In the paragraph starting on line 794, the authors noted that students made connections between two aspects of the same system instead of making analogies between different systems; this suggests that students may not have been making analogies at all, or may not have understood how to create analogies.

My main concern was that this manuscript was written in a non-academic register. Phrasings such as "variables that really matter," "a good majority of the children," and "certainly successful" are examples of informal registers. Academic registers are more formal. 

Minor concerns include the use of incorrect prepositions and the misspelling of multiple words, both occurring multiple times throughout the manuscript.

Author Response

See the uploaded file

Reviewer 2 Report

I would like to start by thanking the author for this very interesting perspective on circuits and education.  Overall, the paper is well-written and well-organized.  I found the motivation and the discussion of the underlying common theme of circuits to be an especially powerful framing for the work.  The theoretical underpinnings are very broad and thorough, skillfully framing the research described later.  The interventions and their experiments are well described and the limitations of the interventions are thoroughly discussed.  I have only minor comments on wording and references (see below), and I strongly support publication of the article.

Line 64 the first sentence is unclear

Line 75 “lit” should be “lights”

I don’t know whether it is necessary, but there has also been research published on electric circuits at the university level.

Line 224 “included” should be “including”

Line 459 “in” should be “of”

Line 499 “for should be “with”

Line 675 is missing a verb, probably “was” before “because”

No additional comments

Round 2

Round 3

Reviewer 1 Report

I removed my first two feedbacks from this document to help make it easier to read. The authors may need to refer to the previous feedback document for context.

“Speaking of” has been reformulated as “regarding”.

“impossibility of referring to mathematics” has been reformulated as “impossibility of using mathematics”.

Choral: corrected, thanks.

Okay. I only listed a few examplesthere are more throughout the manuscript.

On the suggestion to always also report the number of children in addition to the percentage:

it seems to us that this option is not necessary (it is always possible to calculate the number from the percentage anyway) and would on the other hand unnecessarily burden the text.

Reporting the number of children in addition to the attendance provides transparency. While readers can calculate the number from the percentage, it is inconvenient. Adding (N=8), for example, after a percentage would not necessarily burden the text.

Quote: we do not see in the Instructions for Authors of this journal any specific indication of indenting. Nevertheless, we have now followed the suggestion by the referee and have indented this block quote.

Okay.

More on this quote. The reviewer’s question is “how does it relate to abstraction?” We kindly ask the reviewer to read the sentence one more time: it clearly describes a process of abstraction. In particular, the passage 'Perhaps familiarity with the concept of speed prevents you from appreciating this experience of creating order out of a chaos of sense impressions by abstracting some measurable data from it' expressly recurs to the term ‘abstracting’.

The authors have misinterpreted my feedback. It was not about the content written in the block quote, but about the transitions between paragraphs (please refer to my original feedback in the previous version).

It was not clear what the authors want readers to take away from reading the block quote with respect to the authors study (which seems to focus on abstraction) and how this quote connects to the following paragraphs.

We understand the reviewer's invitation to spend further words and examples on the concept of abstraction, but it seems to us that the words already spent describing this topic (as well as those on analogy and image-schematics) are more than enough, perhaps even too many for our article, which is certainly not intended as a review article on such topics.

I point out instances in the manuscript where readers may have difficulties understanding the study.

We partially take up the suggestion about the use of too many synonyms. We have reduced now the variety of terms with which children are described in the text. However, we have left a minimum of variety, because we believe that a text, however academic, with terms that are too repetitive can be excessively boring and that there is no possibility of confusing the reader in any case. Moreover, these words are not always synonyms that can be used indifferently. That is why next to 'pupils' and 'children' we have sometimes left the expression 'students' (which also includes older children).

Okay.
